# PF4 Autoantibody Complexes Cause Activation of Integrins αIIbβ3 and αvβ3 and Possible Subsequent Thrombosis and Autoimmune Diseases

**DOI:** 10.3390/ijms262110260

**Published:** 2025-10-22

**Authors:** Yoko K. Takada, Chun-Yi Wu, Yoshikazu Takada

**Affiliations:** 1Department of Dermatology, University of California Davis School of Medicine, Sacramento, CA 95817, USA; 2Department of Neurology, University of California Davis School of Medicine, Sacramento, CA 95817, USA; chywu@health.ucdavis.edu; 3Department of Biochemistry and Molecular Medicine, University of California Davis School of Medicine, Sacramento, CA 95817, USA

**Keywords:** PF4, allosteric integrin activation, αvβ3, αIIbβ3, site 2, anti-PF4 autoantibody

## Abstract

Previous studies suggest that multiple inflammatory chemokines (e.g., CCL5, CXCL12) bind to the allosteric site of integrins (site 2) and induce allosteric integrin activation and inflammatory signals. PF4 is abundantly present in platelet granules, but PF4 levels are very low in plasma. PF4 is released from damaged platelets and is markedly increased in plasma (>1000×) in pathological conditions. PF4 (tetramer) is an inhibitory chemokine, and the specifics of PF4 signaling are unclear. Docking simulation predicted that PF4 monomer binds to site 2, but PF4 by itself did not induce allosteric integrin activation. Anti-PF4 mAbs KKO and RTO generate complexes with PF4 tetramer and monomer, respectively. We discovered that the PF4/RTO complex induced potent integrin activation, but the PF4/KKO complex did not. We hypothesize that inactive PF4 tetramer is converted by RTO to active monomer. A PF4 mutant (4E), in which four basic amino acid residues in the predicted site 2 binding site were mutated to Glu, did not induce integrin activation and acted as a dominant-negative antagonist, suggesting that the RTO/PF4 complex is required to bind to site 2 for integrin activation. Notably, RTO-like autoantibody was detected in plasma of healthy people. We propose that autoanti-PF4 in healthy controls may not be a problem since plasma PF4 levels are very low. When plasma PF4 tetramer is increased, active PF4 monomer is generated by autoanti-PF4 and plays a role in disease pathogenesis. Notably, anti-inflammatory cytokine neuregulin-1 and anti-inflammatory ivermectin bind to site 2 and suppress integrin activation induced by RTO/PF4 complex, suggesting that neuregulin-1 and ivermectin are potentially useful to suppress PF4/anti-PF4-mediated inflammatory signals.

## 1. Introduction

Integrins are a superfamily of αβ heterodimers that were originally identified as receptors for extracellular matrix proteins [1]. We previously discovered that the chemokine domain of pro-inflammatory chemokine CX3CL1 is a ligand for integrins αvβ3 and α4β1 and binds to the classical ligand-binding site of integrins (site 1) [2]. We showed that CX3CL1 activated soluble integrin αvβ3 in cell-free conditions in an ELISA-type activation assay [3]. In this assay, we coated the wells of 96-well plate with specific integrin ligands and incubated with soluble integrins in the presence of CX3CL1 in 1 mM Ca^2+^ to keep the integrin inactive. We detected an increase in integrin binding to immobilized ligand γC399tr, a fibrinogen fragment, indicating that soluble integrins were activated by CX3CL1. CX3CL1 binds to the allosteric site (site 2) in the integrin headpiece, which is distinct from the classical ligand-binding site (site 1) [3]. Site 2 is located on the opposite side of site 1 in the integrin headpiece. Site 2 was identified by docking simulation of the interaction between the closed/inactive integrin αvβ3 (1JV2.pdb) and CX3CL1 [3]. Other inflammatory chemokines such as stromal-cell-derived factor-1 (SDF-1, CXCL12) and Rantes (CCL5) activated integrins αvβ3 and αIIbβ3 by binding to site 2 [4,5]. Also, a major inflammatory cytokine TNF was shown to bind to site 2 and induce allosteric activation of integrins [6]. 25-Hydroxycholesterol, a mediator of inflammatory signals in innate immunity, was shown to bind to integrin site 2 and induce integrin activation and inflammatory signaling, leading to over-production of inflammatory cytokines (e.g., IL-6 and TNFα) in monocytes [7]. It has thus been proposed that site 2 plays a critical role in integrin activation by inflammatory chemokines and inflammatory signaling.

Previous studies showed that anti-inflammatory cytokine FGF1 and NRG1 bind to site 2 and inhibit integrin activation by inflammatory cytokines [8,9]. Most recently, we discovered that ivermectin (IVM), an anti-inflammatory agent, binds to site 2 and inhibits integrin activation by inflammatory cytokines [6]. These findings suggest that site 2 is critically involved in inflammatory signaling and is a potential therapeutic target.

PF4 (CXCL4) is a CXC-chemokine and is predominantly synthesized in megakaryocytes, sequestered in platelet-granules and released upon platelet activation [10]. PF4 is known to inhibit inflammation, angiogenesis, and tumor growth, although the mechanisms of PF4’s actions have not been fully elucidated [11,12,13,14,15,16,17,18,19]. PF4 is known to bind to αvβ3 [10] and Mac-1 [13]. It is unclear if PF4 binding to integrins is involved in PF4-mediated diseases. It is also unclear if PF4 binds to αIIbβ3 and activates it. The goal of the present study is to elucidate the role of integrins in PF4-mediated diseases. We performed a docking simulation of the interaction between PF4 and the closed/inactive headpiece αvβ3. The simulation predicted that PF4 binds to site 2, but PF4 did not activate integrins. We hypothesized that PF4 binding to site 2 is critically involved in the pathogenesis of diseases. We discovered that the PF4/monoclonal antibody specific to PF4 (RTO) complex potently induced activation of integrins. Also, RTO-like autoanti-PF4 is present in plasma from healthy controls. The PF4 (4E) mutant defective in site 2 binding due to mutations in the predicted site-2-binding interface was defective in integrin activation and acted as a dominant-negative antagonist. The PF4 (4E) mutant may have therapeutic potential.

## 2. Results

### 2.1. PF4 Specifically Binds to Site 1 and Site 2 of Integrin αvβ3

Although PF4 is known to bind to integrin αvβ3, it is unclear how PF4 binds to integrin αvβ3. To predict how PF4 binds to integrin αvβ3, we performed a docking simulation of interaction between PF4 (1RHP.pdb) and integrin αvβ3 using autodock3. In our docking studies, PF4 is predicted to bind to site 1 (docking energy −24.3 kcal/mol) of the active conformer of integrin αvβ3 (open headpiece/active, 1L5G.pdb) (Figure 1a). 25-Hydroxycholesterol (25HC), a major inflammatory mediator, is known to bind to the allosteric site (site 2) and induce integrin activation and inflammatory signals. We showed that several inflammatory chemokines bound to the allosteric site (site 2) and activated integrins. PF4 is known to inhibit angiogenesis and tumor growth, but whether inhibitory PF4 binds to site 2 remains unclear. Docking simulation of the interaction between PF4 (1RHP.pdb) and an inactive conformer of integrin αvβ3 (closed headpiece/inactive 1JV2.pdb) predicted that PF4 binds to site 2 (docking energy −21.94 kcal/mol) (Figure 1b), predicting that PF4 allosterically activates integrins as in CX3CL1, CCL5, or CXCL12. When Figure 1a,b are superposed, PF4 can bind to two distinct binding sites (site 1 and site 2) (Figure 1c).

### 2.2. PF4 Binds to Site 1 of Activated Soluble Integrins αvβ3 and αIIbβ3

PF4 is stored in platelet granules and released from platelets upon platelet degradation or activation [10]. We previously showed that several cytokines that are stored in platelet granules bind to and activate αIIbβ3 [5]. This leads to platelet activation and thrombosis. We predicted that PF4 binds to αIIbβ3 and thereby induces platelet activation and thrombosis. Consistent with the prediction, we showed that PF4 bound to soluble αvβ3 and αIIbβ3, which are activated by 1 mM Mn^2+^ in ELISA-type binding assays in cell-free conditions in a dose-dependent manner (Figure 2a). To confirm that the binding of PF4 to soluble integrins is not due to an abnormality of our PF4 preparations, we tested if authentic PF4 (Invitrogen) behaves similarly to PF4 that was generated in our lab. We obtained similar results using authentic PF4 (Figure 2b), indicating that the binding of PF4 to integrins is a property of PF4. All experiments were thus performed using PF4 preparations generated in our lab if not indicated otherwise.

We found that the disintegrin domain of ADAM15, which is known to bind to integrins αvβ3 [20] and αIIbβ3 [21], inhibited the binding of PF4 to integrins, but control GST did not (Figure 2c,d). This indicates that the PF4 binding to these integrins is specific.

**Figure 2 ijms-26-10260-f002:**
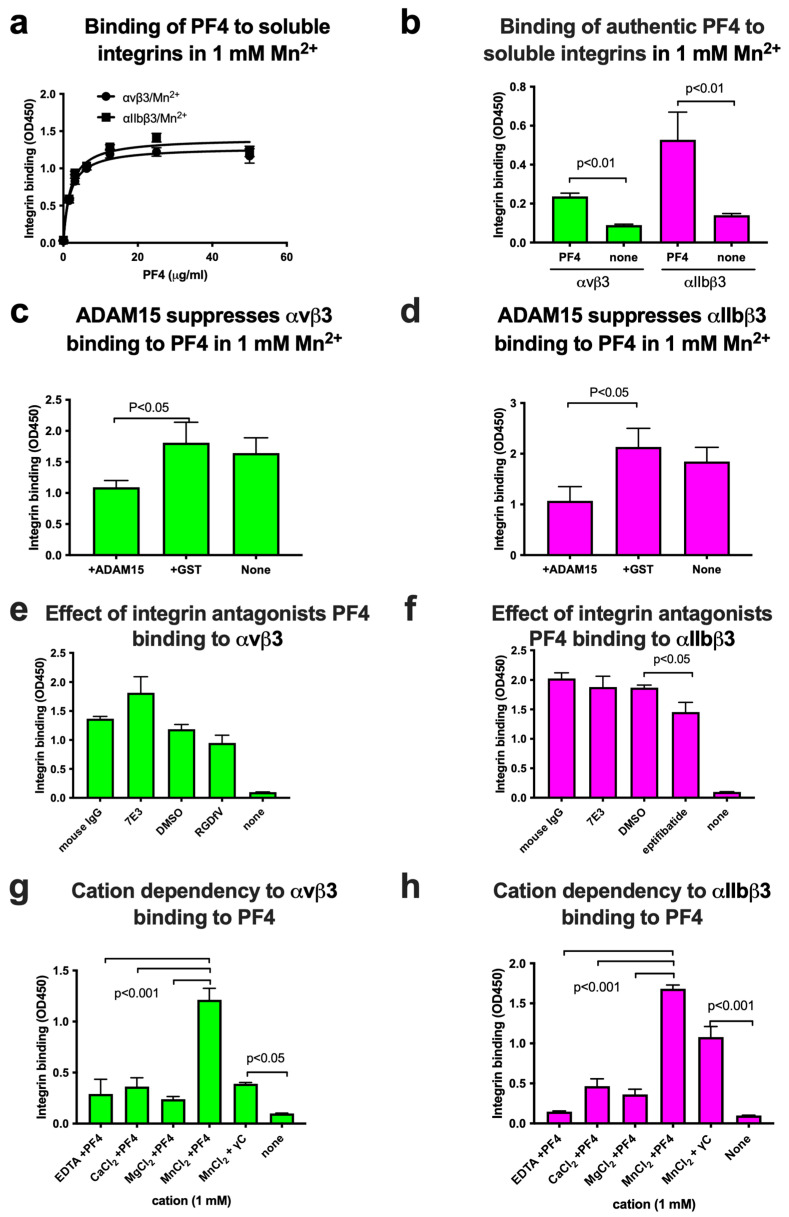
PF4 specifically binds to soluble αIIbβ3 and αvβ3 in ELISA-type binding assays in 1 mM Mn^2+^. (**a**) PF4 binds to soluble integrins in 1 mM Mn^2+^ in ELISA-type binding assays. PF4 was immobilized to wells of a 96-well microtiter plate and incubated with soluble αIIbβ3 or αvβ3 (1 μg/mL) in Tyrode–HEPES buffer with 1 mM Mn^2+^ (to activate integrins) for 1 h at room temperature, and bound integrins were quantified using anti-β3 (mAb AV10) and HRP-conjugated anti-mouse IgG. Data is shown as means +/− SD in triplicate experiments. The data show that PF4 binds to these integrins at Kd < 1 μg/mL. (**b**) Binding of authentic PF4 (Invitrogen) to integrins. Binding assays were performed as described in (**a**). PF4 (6.25 μg/mL) was used. Data is shown as means +/− SD in triplicate experiments. The data show that PF4 binding to soluble integrins is not due to the source of PF4. (**c**,**d**) The binding of PF4 to integrins was suppressed by the disintegrin domain of ADAM15 fused to GST (ADAM15 disintegrin) but not by control GST. To establish the specificity of PF4 binding to soluble integrins αIIbβ3 (**b**) and αvβ3 (**c**), we studied if ADAM15 disintegrin, which is known to bind to integrins αIIbβ3 [21] and αvβ3 [20], suppresses the binding. ADAM15 disintegrin (100 μg/mL) suppressed the integrin binding to immobilized PF4 (12.5 μg/mL), but control GST (100 μg/mL) did not. Data is shown as means +/− SD in triplicate experiments. This indicates that the binding of soluble integrins to PF4 is specific. (**e**,**f**) Effect of antagonists to integrins on PF4 binding. Wells of 96-well microtiter plates were coated with PF4 (50 μg/mL) and incubated with soluble integrins (1 μg/mL) in Tyrode–HEPES buffer with 1 mM Mn^2+^. mAb 7E3 (specific to human β3, 10 μg/mL), eptifibatide (specific to αIIbβ3, 0.65 μg/mL), or cyclic RGDfV (specific to αvβ3, 10 μM) were mixed with soluble integrins first and then added to the wells. Assays were performed as described in (**a**). Data is shown as means +/− SD in triplicate experiments. (**g**,**h**) Cation dependency of PF4 binding to integrins. Wells of 96-well microtiter plates were coated with PF4 (50 μg/mL) or full-length γC (50 μg/mL), which binds to αIIbβ3 and αvβ3, as a positive control. Wells were incubated with soluble integrins (1 μg/mL) in Tyrode–HEPES buffer with different cations (1 mM). Bound integrins were quantified as described in (**a**). Data is shown as means +/− SD in triplicate experiments. None in (**e**–**h**) do not contain PF4.

However, known antagonists for these integrins did not effectively inhibit PF4 binding to these integrins, except that eptifibatide (specific to αIIbβ3) weakly inhibited (Figure 2e,f). Mn^2+^ (1 mM) supported PF4 binding to site 1 well, but other cations were less active. Also, the full-length fibrinogen C-terminal domain (γC151-411), a ligand for αvβ3 and αIIbβ3, was used as a positive control (Figure 2g,h).

### 2.3. PF4 Is Predicted to Bind to Site 2 but Did Not Activate Integrins

Previous studies showed that inflammatory chemokines CX3CL1, CXCL12, and CCL5 bound to site 2 and activated integrins αIIbβ3 and αvβ3 [2,3,5]. Although PF4 is predicted to bind to site 2, PF4 by itself did not activate αIIbβ3 or αvβ3 (Figure 3a,b). This contrasts with our previous findings that inflammatory chemokines potently activated these integrins by binding to site 2. We hypothesize that the inhibitory actions of PF4 may be related to its inability to activate integrins and subsequent inflammatory signaling upon binding to site 2.

### 2.4. Anti-PF4 (RTO) Changes the Phenotype of PF4 from Inhibitory to Activating

Anti-PF4 autoantibodies have been detected in a vaccine-induced catastrophic thrombotic thrombocytopenia (VITT) disorder in vaccination for SARS-CoV-2, and this disorder presents as extensive thrombosis in atypical sites, primarily in the cerebral venous, alongside thrombocytopenia. Anti-PF4 is also detected in several autoimmune diseases (e.g., SLE, systemic sclerosis, and RA) [12,22,23].

A murine anti-hPF4 mAb RTO is specific to PF4 and does not require heparin for binding. Murine mAb KKO binds specifically to hPF4/heparin complexes but not to PF4 [24]. We studied if RTO and KKO influence the ability of PF4 to induce integrin activation. Notably, the RTO (at 10 μg/mL)/PF4 complex markedly activated integrins αvβ3 and αIIbβ3 in ELISA-type activation assays in 1 mM Ca^2+^ at physiological concentrations of PF4 (<1 μg/mL). PF4 itself, KKO/PF4 complex, or mouse IgG did not activate integrins (Figure 3a,b).

We obtained similar results using authentic PF4 (Invitrogen) instead of PF4 produced in our laboratory (Figure 3c,d), indicating that the potent activation of integrins by RTO/PF4 complex is a property of PF4.

This is the first evidence that anti-PF4 drastically changed the phenotype of PF4 upon binding to site 2 and induced integrin activation (Figure 3e). Since heparin-independent anti-PF4 has been detected in autoimmune diseases and the levels of anti-PF4 correlate with disease progression [2,22], activation of integrins by anti-PF4/PF4 complex may be involved in the pathogenesis of autoimmune diseases.

We superposed the PF4/RTO complex (1RHP.pdb) and PF4/αvβ3 complex (Figure 3f). It is predicted that RTO, PF4, and integrin can co-exist without steric hindrance. The predicted RTO-binding site and site-2-binding site in PF4 are distinct.

### 2.5. PF4 Mutant Defective in Site 2 Binding Is Defective in Integrin Activation and Acted as an Antagonist for PF4/RTO-Induced Integrin Activation

Table 1 shows amino acid residues in PF4 that are involved in site 2 binding (Figure 1b). To determine if PF4/RTO complex activates integrins by binding to site 2, we developed PF4 mutants that are defective in site 2 binding by introducing mutations in the site-2-binding interface in PF4 predicted by docking simulation. Arg20, Arg22, Lys46, Arg49, Lys65, and Lys66 in the integrin-binding interface of PF4 were mutated to Glu. Positions of these amino acid residues are shown in Figure 4a. The R20E/R22E and K46E/R49E mutations showed reduced ability to mediate RTO/PF4-induced integrin activation (Figure 4b,c). The combined PF4 mutant (R20E/R22E/K46E/R49E) most effectively reduced RTO/PF4-induced integrin activation (Figure 4b,c). Notably, this mutant suppressed integrin activation induced by PF4/RTO complex in a dose-dependent manner (dominant-negative effect) (Figure 4d,e). This PF4 mutant binds to anti-PF4 (RTO) but cannot induce integrin activation since it cannot bind to site 2. The PF4 mutant competes with wild-type PF4 complex for binding to anti-PF4.

### 2.6. Plasma from Healthy Controls Contains Antibodies That Activate Soluble Integrins in a PF4-Dependent Manner but Sera from SLE or RA Patients Did Not

We hypothesized that sera from patients with autoimmune diseases contain autoantibodies to PF4 that activate integrins in the presence of wt PF4. However, we did not detect autoantibodies to PF4 in SLE or RA patient sera or sera from healthy controls (Figure 5a). Instead, we detected RTO-like anti-PF4 in plasma from SLE patients and healthy controls but not sera. We found that plasma from an SLE patient (#70) activates soluble integrin αvβ3 with WT PF4 but not with the PF4 4E mutant, suggesting that anti-PF4 in this plasma acts like RTO. These findings suggest that the PF4/anti-PF4 from human plasma requires site 2 binding to induce integrin activation.

To further identify autoanti-PF4 in human plasma, we affinity-purified autoanti-PF4 from plasma from healthy controls (pooled plasma) using PF4-affinity resin. We detected bands with mol. wt 150K (Appendix A), suggesting that the antibody is IgG. Mass spectroscopy detected 25 heavy-chain variable domains and light-chain variable domains (19 kappa and 20 lambda), suggesting that the purified antibodies are a mixture of IgG. We repeated purification using a single healthy individual, but again, we obtained multiple IgG heavy and light chains, suggesting that anti-PF4 antibodies are polyclonal (Appendix A). We could not obtain sequences of hypervariable domains and reconstruct the recombinant IgG.

### 2.7. Ivermectin Inhibits RTO/PF4-Mediated Integrin Activation

We recently discovered that anti-inflammatory cytokine neuregulin-1 (NRG1) binds to site 2 and suppresses integrin activation by multiple inflammatory cytokines [2,3,4,5,6]. We superposed the predicted binding site for PF4 monomer and that of NRG1 overlap (Figure 6a). We also discovered that a potential anti-inflammatory agent, ivermectin (IVM), binds to site 2 and inhibits allosteric integrin activation by pro-inflammatory cytokines (e.g., TNF and CCL5) [6]. The predicted binding site for PF4 monomer and that of IVM overlap (Figure 6b). We found that NRG1 inhibited integrin activation by the PF4/RTO complex (Figure 6c). Also, IVM inhibited PF4/RTO-mediated integrin activation (Figure 6d). These findings suggest that NRG1 and IVM are potential antagonists for PF4/autoanti-PF4 induced autoimmune diseases.

## 3. Discussion

Multiple inflammatory chemokines (CCL5, CXCL12, and CX3CL1) bind to site 2 and potently activate integrins. PF4 is known to be a tetramer. In the present study, docking simulation using monomeric PF4 predicted that PF4 binds to the allosteric site (site 2) of integrins. We discovered that PF4 in complex with RTO activates integrins, but PF4 without RTO does not activate integrins. PF4 in RTO/PF4 complex is monomeric, and thus, we suspect that RTO breaks tetramer PF4 into monomer PF4, and thereby, monomeric PF4/RTO complex binds to site 2 and induces integrin activation. The PF4 mutant in the predicted site-2-binding interface was defective in integrin activation in the presence of RTO. This indicates that PF4/RTO complex binding to site 2 is critical for PF4-induced integrin activation. Also, this PF4 mutant acted as an antagonist for PF4/anti-PF4-induced integrin activation and has potential as an antagonist for PF4-induced autoimmune diseases. Anti-PF4 KKO is known to bind to PF4 tetramer but not PF4 monomer, although KKO induces heparin-induced thrombocytopenia (HIT). RTO/PF4 monomer complex induces integrin activation. Therefore, the present RTO-induced integrin activation and subsequent autoimmune disease and KKO-induced HIT are distinct.

We hypothesized that autoanti-PF4 antibody is present in sera of patients with autoimmune diseases (e.g., SLE and RA), and such autoanti-PF4 induces autoimmune diseases. Unexpectedly, we did not detect RTO-like anti-PF4 in sera from patients but in plasma of healthy controls. It is known that PF4 levels in circulation are very low in healthy individuals, and thus, the presence of RTO-like anti-PF4 may not be a problem in healthy controls. Plasma PF4 tetramer levels are markedly enhanced when platelets are damaged or stimulated. We propose that plasma PF4 monomer levels are critical for pathogenesis of the diseases. Once PF4 tetramer levels increase, autoanti-PF4 will generate PF4 monomers, which may contribute to disease pathogenesis. Autoanti-PF4 antibodies from healthy controls were purified using PF4-affinity chromatography and analyzed by mass-spectroscopic sequencing. It appears that autoanti-PF4 antibodies are polyclonal. Thus, it is difficult to obtain more detailed information (e.g., sequences of hypervariable domains of IgG) and reconstruct autoanti-PF4 at this point. We found that PF4 activation can be blocked by PF4 mutant (4E) defective in site 2 binding, indicating that activation of PF4 by autoantibodies from control plasma resembles that induced by mAb RTO, consistent with the notion that mAb RTO functions similarly to human autoanti-PF4. We discovered that anti-inflammatory NRG1 and IVM suppressed PF4/anti-PF4-induced integrin activation, suggesting that binding of PF4 to site 2 may be a common pathway of inflammatory signaling. These findings suggest that it is possible to block anti-PF4/PF4-induced inflammatory signaling by using these antagonists (PF4 mutant 4E, NRG1, and IVM). We will study if similar PF4 activation mechanisms are present in animals. Although PF4 levels are very low in normal plasma, when platelets are degraded or activated, PF4 is released from platelet granules, and plasma PF4 levels dramatically increase (×1000). PF4 is activated by autoanti-PF4 in plasma (PF4 tetramer is possibly converted to monomer), and PF4/anti-PF4 binds to site 2 of platelet αIIbβ3 and activates this integrin. Activated αIIbβ3 binds to plasma fibrinogen and induces platelet aggregation. We propose that this may be a potential mechanism of PF4/anti-PF4-mediated thrombosis.

## 4. Experimental Procedures

### 4.1. Fibrinogen γ-Chain C-Terminal Residues 390-411, a Specific Ligand for αIIbβ3 Fused to GST

cDNA encoding (6 His tag and fibrinogen γ-chain C-terminal residues 390-411) [HHHHHH]NRLTIGEGQQHHLGGAKQAGDV] was conjugated with the C-terminus of GST (designated γC390-411) in pGEXT2 vector (BamHI/EcoRI site). The protein was synthesized in *E. coli* BL21 and purified using glutathione affinity chromatography.

### 4.2. The Fibrinogen γ-Chain C-Terminal Domain (γC399tr, Residues 151-399)

A specific ligand for αvβ3 and full-length γC (residues 151-411), which binds to both αvβ3 and αIIbβ3, have been previously described [25]. The disintegrin domain of ADAM15 fused to GST (ADMA15 disintegrin) and parent GST were synthesized as previously described [20]

### 4.3. PF4

The cDNA encoding PF4 was synthesized and subcloned into the BamHI/EcoRI site of pET28a vector. Protein synthesis was induced by IPTG in *E. coli* BL21, and protein was synthesized as insoluble inclusion bodies and purified in Ni-NTA affinity chromatography under denaturing conditions and renatured as described [8]. PF4 authentic control and anti-PF4 antibody RTO and anti-PF4/heparin complex KKO were obtained from Invitrogen.

### 4.4. ELISA-Type Integrin Activation Assays [9]

Wells of 96-well microtiter plates were coated with γC390-411 (a specific ligand for αIIbβ3) or γC399tr (a specific ligand for αvβ3). Remaining protein binding sites were blocked with BSA. Soluble recombinant αIIbβ3 or αvβ3 (AgroBio, 1 μg/mL) was pre-incubated with PF4 or anti-PF4/PF4 for 10 min at room temperature and was added to the wells and incubated in HEPES–Tyrodes buffer with 1 mM CaCl_2_ for 1 h at room temperature. After unbound αIIbβ3 or αvβ3 was removed by rinsing the wells with binding buffer, bound αIIbβ3 or αvβ3 was measured using anti-integrin β3 mAb (AV-10) followed by HRP-conjugated goat anti-mouse IgG and peroxidase substrates.

### 4.5. ELISA-Type Integrin Binding Assays [3]

Wells of 96-well microtiter plates were coated with PF4. Remaining protein binding sites were blocked by incubating with BSA. After washing with PBS, soluble recombinant αIIbβ3 or αvβ3 (1 μg/mL) was added to the wells and incubated in HEPES–Tyrodes buffer with 1 mM MnCl_2_ for 1 h at room temperature. Bound αIIbβ3 or αvβ3 was measured using anti-integrin β3 mAb (AV-10) followed by HRP-conjugated goat anti-mouse IgG and peroxidase substrates.

### 4.6. Docking Simulation

Docking simulation of the interaction between PF4 and integrin αvβ3 (open-headpiece form, 1L5G, or closed-headpiece form, PDB code 1JV2) was performed using AutoDock3 as described previously [26]. We used the headpiece (residues 1–438 of αv and residues 55–432 of β3) of αvβ3. Cations were not present in integrins during docking simulation. The ligand was presently compiled to a maximum size of 1024 atoms. Atomic solvation parameters and fractional volumes were assigned to the protein atoms by using the AddSol utility, and grid maps were calculated by using the AutoGrid utility in AutoDock 3.05. A grid map with 127 × 127 × 127 points and a grid point spacing of 0.603 Å included the headpiece of αvβ3. Kollman ‘united-atom’ charges were used. AutoDock 3.05 uses a Lamarckian genetic algorithm (LGA) that couples a typical Darwinian genetic algorithm for global searching with the Solis and Wets algorithm for local searching. The LGA parameters were defined as follows: the initial population of random individuals had a size of 50 individuals; each docking was terminated with a maximum number of 1 × 10^6^ energy evaluations or a maximum number of 27,000 generations, whichever came first; and mutation and crossover rates were set at 0.02 and 0.80, respectively. An elitism value of 1 was applied, which ensured that the top-ranked individual in the population always survived into the next generation. A maximum of 300 iterations per local search were used. The probability of performing a local search on an individual was 0.06, whereas the maximum number of consecutive successes or failures before doubling or halving the search step size was 4.

### 4.7. Mass Spec Sequencing

Protein samples were digested via suspension-trap devices (S-Trap) (ProtiFi). S-Trap is a powerful filter-aided sample preparation (FASP) method that consists in trapping acid-aggregated proteins in a quartz filter prior enzymatic proteolysis. In the same tube, we reduced and alkylated the samples, followed by overnight tryptic proteolysis. Peptides were eluted from S-Trap by sequential elution buffers of 100mM TEAB, 0.5% formic acid, 50% acetonitrile, and 0.1% formic acid. The eluted tryptic peptides were dried in a vacuum centrifuge and re-constituted in 0.1% trifluoroacetic acid. These were subjected to LC-MS analysis. The HPLC system was a Thermo Scientific Dionex UltiMate 3000 RSLC system using a PepSep analytical column (PepSep, Nordhavn, Denmark) as follows: 150 µm × 8 cm C18 column with a 1.5 μm particle size (100 Å pores), preceded by a PepSep C18 guard column, and heated to 40 °C. The gradient was 60 min, with mobile phases as follows: A: water/0.1% formic acid; and B: 80%ACN/0.1% formic acid. Eluting peptides were directly sprayed into an Orbitrap Exploris 480 instrument (Thermo Fisher Scientific, Bremen, Germany) for Top30 data-dependent analysis. The full MS resolution was set to 60,000 at m/z 200, and the mass range was set to 350–1500Da. For fragmentation spectra, a resolution of 15,000 was used. The LCMS raw files were processed with Proteome Discoverer 2.5 (Thermo Fisher) using the integrated SEQUEST engine. All data was searched against a target/decoy version of the Homo Sapiens Uniprot reference proteome, UP000005640, without isoforms (21,074 entries), downloaded on January 2025 along with a database of 381 common laboratory protein contaminants (https://github.com/HaoGroup-ProtContLib accessed on 15 May 2025). Peptide tolerance was set to 10 ppm, and fragment mass tolerance was set to 0.6 Da. Trypsin was specified as the enzyme, cleaving after all lysine and arginine residues and allowing up to two missed cleavages. Carbamidomethylation of cysteine was specified as a fixed modification, and protein N-terminal acetylation, oxidation of methionine, deamidation of asparagine and glutamine, and pyro-glutamate formation from glutamine were considered variable modifications with a total of 2 variable modifications per peptide. The false discovery rate (FDR) was set to 1% on peptide spectrum match (PSM), PTM site, and protein level.

### 4.8. Other Methods

Treatment differences were tested using ANOVA and Tukey’s multiple comparison test to control the global type I error using Prism 10 (Graphpad software).

## 5. Conclusions

We performed a docking simulation of the interaction between PF4 and the closed/inactive headpiece αvβ3. The simulation predicted that PF4 binds to site 2, but PF4 did not activate integrins in integrin activation assays. This contrasts with inflammatory chemokines (CCL5, CXCL12, and CX3CL1), which bind to site 2 and activate integrins. It is unclear why PF4 cannot activate integrins, although PF4 binds to site 2. We discovered that the PF4/monoclonal antibody specific to PF4 (RTO) complex potently induced activation of integrins. PF4 mutants, in which the predicted site-2-binding interface was mutated, were defective in site 2 binding and did not induce integrin activation and acted as an antagonist of PF4/RTO-induced integrin activation. These findings suggest that anti-PF4 changed the phenotype of PF4 from inhibitory to inflammatory. We propose that PF4 tetramer is inactive but PF4 monomer/RTO complex is active. Also, anti-PF4 may be involved in the pathogenesis of autoimmune diseases by activation of αvβ3 and other integrins in non-platelet cells (e.g., monocytes). The PF4 mutant defective in binding to site 2 is a potential antagonist of anti-PF4-induced VITT and autoimmune diseases. Notably, we discovered that plasma from healthy people contains RTO-like autoanti-PF4, suggesting that PF4 autoantibody may not induce an inflammatory response since plasma PF4 tetramer levels are low in healthy people, but once PF4 tetramer levels are high (e.g., in platelet degradation), PF4 monomer is generated and induces inflammatory responses.

## Figures and Tables

**Figure 1 ijms-26-10260-f001:**
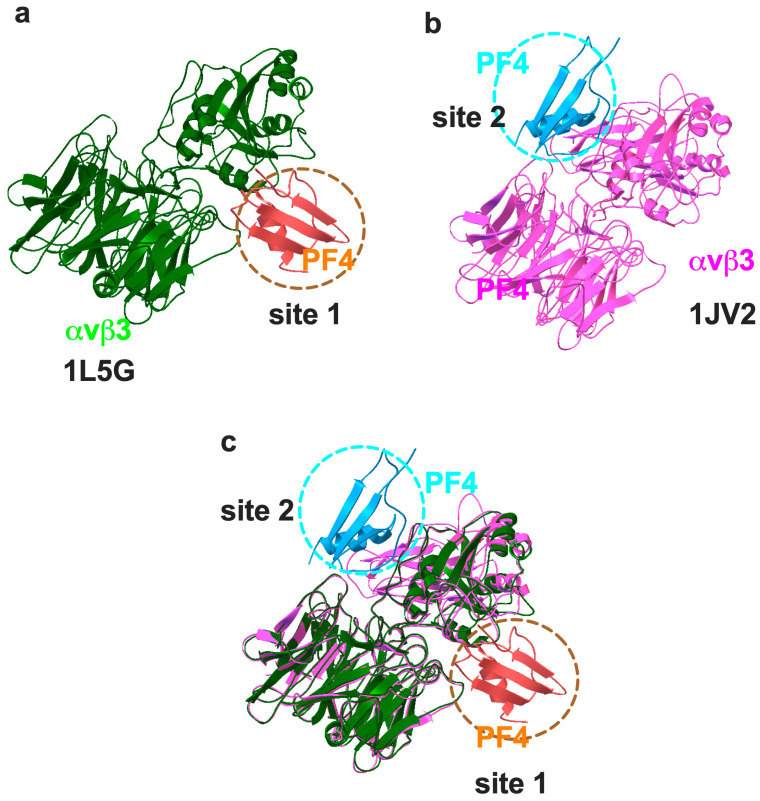
Docking models of anti-PF4/PF4–integrin interaction. (**a**) PF4 binding to the classical ligand-binding site (site 1) of active αvβ3 (1L5G.pdb). Three-dimensional structure of αvβ3 was used because active and inactive three-dimensional structures are known. Autodock3 was used for docking simulation. The simulation predicts that PF4 binds to site 1 (docking energy −24.3 kcal/mol). We showed that this is really the case (Figure 2). (**b**) PF4 binding to the allosteric site (site 2) of inactive αvβ3 (1JV2.pdb). Docking energy −21.94 kcal/mol. Docking models in (**a**,**b**) were superposed (**c**). We hypothesize that PF4 binds to site 2 of inactive integrins but does not induce activation at biological concentrations of PF4 (Figure 3 and Figure 4). We introduced mutations in the predicted site-2-binding interface of PF4.

**Figure 3 ijms-26-10260-f003:**
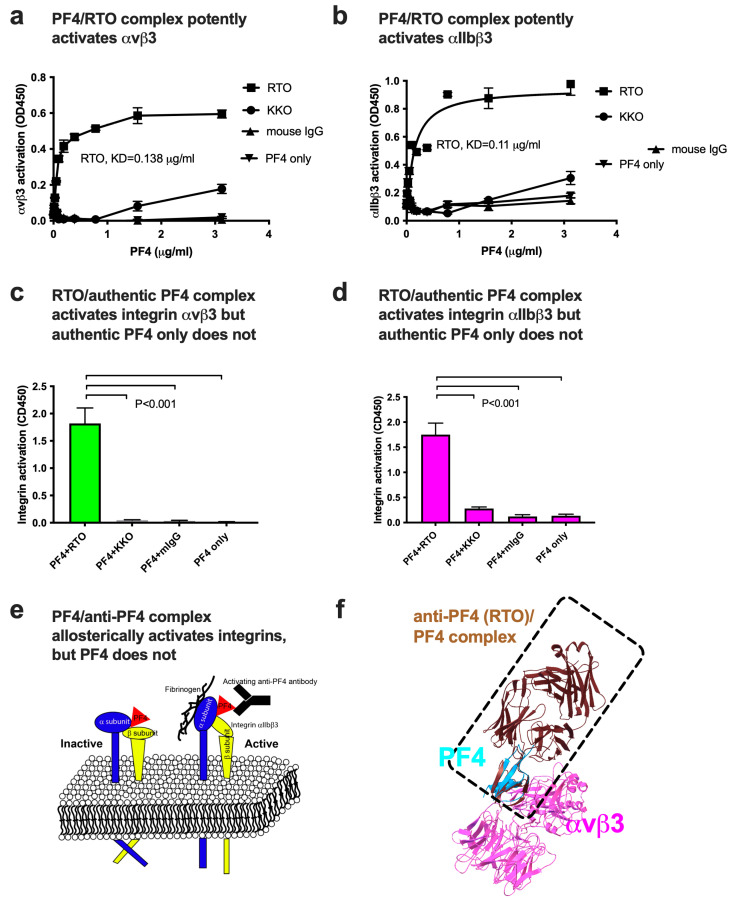
Anti-PF4/PF4 complex potently activated integrins αvβ3 and αIIbβ3, but PF4 itself did not. (**a**,**b**) Anti-PF4 (RTO) markedly enhances PF4-induced activation of soluble αvβ3 (**a**) and αIIbβ3 (**b**) in 1 mM Ca^2+^ in ELISA-type activation assays. The fibrinogen fragments γC390-411 (a specific ligand for αvβ3) was immobilized to wells of a 96-well microtiter plate. Wells were incubated with soluble αvβ3 (1 μg/mL) in Tyrode–HEPES buffer with PF4 in 1 mM Ca^2+^ (to keep integrins inactive) for 1 hr at room temperature. Anti-PF4 (RTO or KKO, 10 μg/mL) was added (without heparin). After washing, bound integrins were quantified using anti-β3 (mAb AV10) and HRP-conjugated anti-mouse IgG. Mouse IgG (10 μg/mL) and no antibody were used as negative controls. (n = 3). The data show that PF4 itself did not activate these integrins at <1 μg/mL, but the RTO/PF4 complex did. (**c**,**d**) Authentic PF4/anti-PF4 (RTO) complex markedly activated soluble integrins αvβ3 (**c**) and αIIbβ3 (**d**) at 1 μg/mL PF4, but authentic PF4/KKO complex did not. Commercial PF4 was used instead of our own PF4 preparation. Activation assays were performed as described in a. The data show that RTO/authentic PF4 complex activated αvβ3 at 1 μg/mL, but the authentic PF4/KKO complex, PF4 with mouse IgG, and PF4 itself, did not. (**e**) The anti-PF4/PF4 complex induces integrin activation, while PF4 alone does not. Anti-PF4 is detected in thrombocytopenia and other autoimmune diseases, and this activation by anti-PF4/PF4 complex may be potentially involved in the pathogenesis of diseases. Our model also predicts that PF4 mutants defective in binding to site 3 may be defective in inducing integrin activation and potentially act as antagonists. (**f**) When the anti-PF4 (RTO)/PF4 complex structure (4RAU.pdb) was superposed, PF4/anti-PF4 was predicted to bind to αvβ3 (site 2) without steric hindrance.

**Figure 4 ijms-26-10260-f004:**
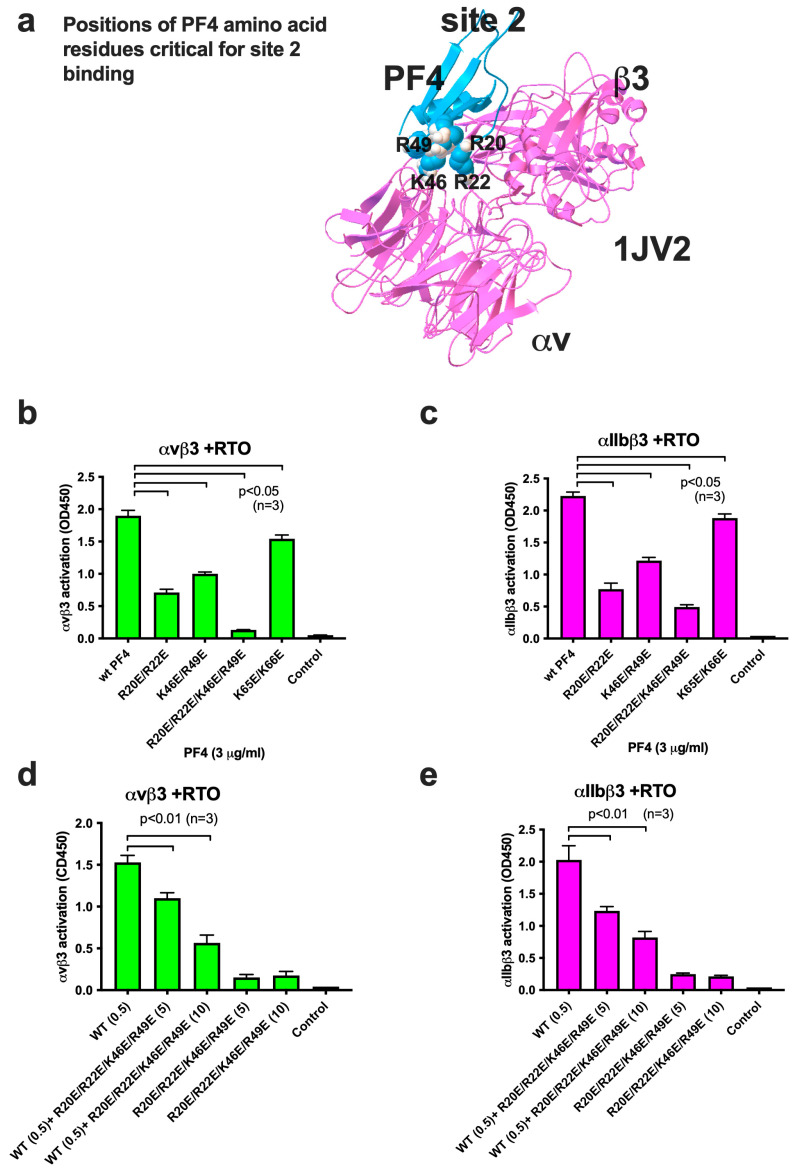
Point mutations of the predicted site-2-binding site of PF4 inhibit integrin activation by PF4/RTO complex, and the PF4 mutant (4E) acts as an antagonist. (**a**) Positions of amino acid residues in the PF4/site 2 interface. (**b**,**c**) Point mutations in the predicted integrin-binding site (site 2) of PF4 suppressed activation of αvβ3 (**b**) and αIIbβ3 (**c**) by the anti-PF4/PF4 complex. The PF4 mutant (R20E/R22E/K46E/R49E) most effectively suppressed integrin activation by anti-PF4/PF4 complex. (**d**,**e**) The R20E/R22E/K46E/R49E mutant suppressed activation of integrin αvβ3 (**d**) and αIIbβ3 (**e**) by the anti-PF4/PF4 complex in a dose-dependent manner. WT PF4 (0.5 μg/mL) and excess PF4 mutant (5 or 10 μg/mL) were used. The data indicates that the R20E/R22E/K46E/R49E mutant acted as an antagonist.

**Figure 5 ijms-26-10260-f005:**
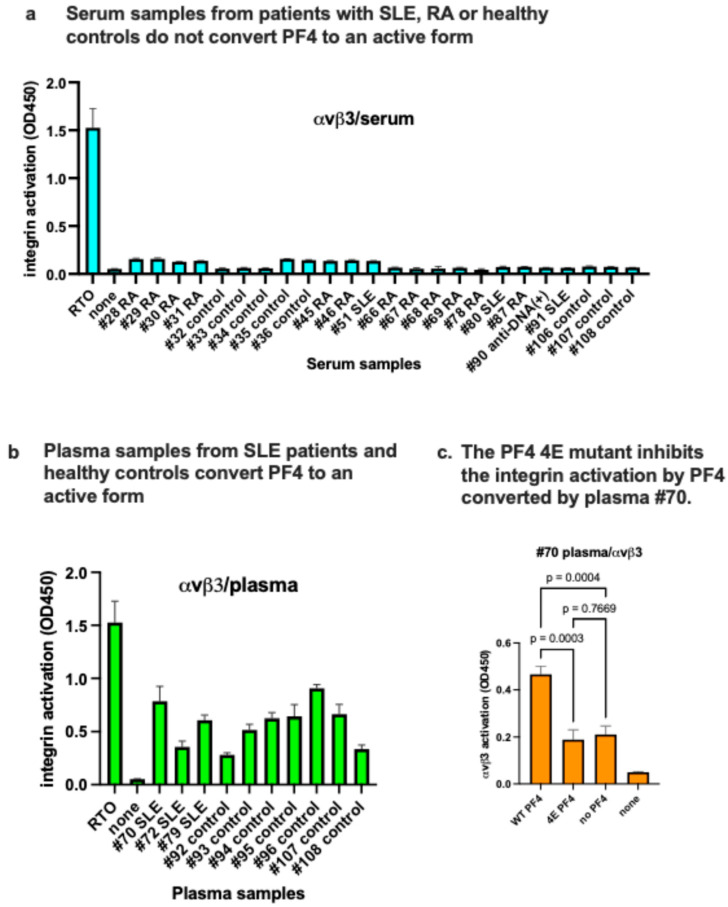
RTO-like anti-PF4 antibodies are present in plasma of SLE and RA patients and healthy controls but not in the sera. Activation of soluble αvβ3 by serum (**a**) or plasma (**b**). Plasma and serum samples were obtained from UC Davis Biorepository. Serum (10 μL) (**a**) and plasma (10 μL) (**b**) and PF4 (1 μg/mL) were incubated with soluble αvβ3, and bound active αvβ3 on immobilized γC399tr was measured, as described above. RTO (1 μg/mL) was used as a positive control. (**c**) The PF4 mutant (4E) inhibits the activation of soluble αvβ3 by PF4 converted by plasma. The 4E mutant (1 μg/mL) was used instead of WT PF4 in integrin activation assays.

**Figure 6 ijms-26-10260-f006:**
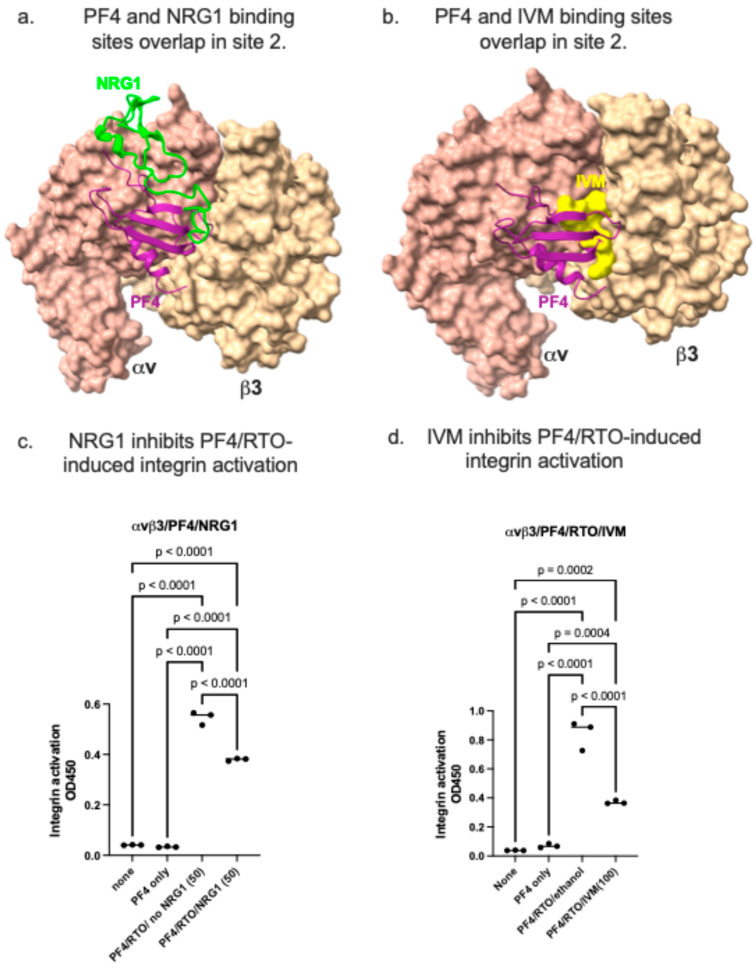
Inhibition of PF4/RTO-mediated integrin activation by anti-inflammatory neuregulin-1 (NRG1) and ivermectin (IVM). (**a**) The predicted binding sites for PF4 and NRG1. (**b**) The predicted binding site for PF4 and IVM. Note that PF4, NRG1, and the IVM-binding site overlap. (**c**) NRG1 inhibits activation of integrin αvβ3 induced by PF4/RTO. Integrin activation was measured via the method in the Experimental Procedures section. (**d**) IVM inhibits activation of integrin αvβ3 induced by PF4/RTO. Integrin activation was measured in the Experimental Procedures section. The data are shown as means +/− SD of triplicate experiments.

**Table 1 ijms-26-10260-t001:** Amino acid residues involved in PF4–integrin αvβ3 site 2 interaction predicted by docking simulation.

PF4 (1RHP.pdb)	αv (1JV2.pdb)	β3 (1JV2.pdb)
Thr15, Thr16, Ser17, Gln18, Val19, **Arg20**, Pro21, **Arg22**, His23, Thr25, **Lys46**, Asn47, **Arg49**, Asp54, Leu55, Gln56, Ala57, Pro58, Leu59, **Lys62**, **Lys65, Lys66**, Leu67, Glu69, Ser70	Glu15, Gly16, Ser17, Tyr18, Phe19, Lys42, Asn44, Ile50, Val51, Glu52, Trp93, Ala396, Ala397, Arg398, Ser399, Met400, Phe427, Gly428, Val429, Asp430	Pro160, Met165, Lys235, Gly264, Ile265, Val266, Gln267, Asp270, Gln272, Cys273, His274, Val275, Gly276, Ser277, Asp278, His280, Tyr281, Ser282, Ala283, Thr285, Thr286, Met287

Amino acid residues within 0.6 nm between PF4 and αvβ3 were selected using PDB viewer (version 4.1). Amino acid residues in PF4 selected for mutagenesis are shown in bold.

## Data Availability

We will share existing datasets or raw data that have been analyzed in the manuscript upon request.

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
