# Peer review of "PF4 Autoantibody Complexes Cause Activation of Integrins αIIbβ3 and αvβ3 and Possible Subsequent Thrombosis and Autoimmune Diseases"

_ijms, 2025, doi:10.3390/ijms262110260_

Round 1
Reviewer 1 Report
Comments and Suggestions for Authors
IJMS-3866123
This study identifies a new mechanism by which anti-PF4/PF4 complexes allosterically activate integrins, providing important insights into the pathogenesis of thrombotic thrombocytopenia and autoimmune diseases. The experiments are well designed, particularly the use of specific antibodies (RTO and KKO) and the site-directed mutant (4E) to examine the activation mechanism. The finding of RTO-like autoantibodies in healthy individuals is interesting and adds clinical relevance. However, several major and minor issues need to be addressed before publication.
Major issues:
- I think the Introduction section is too brief. It lacks background on PF4 as an inhibitory chemokine, which the authors should add. Furthermore, the authors should clearly identify the research gap. Moreover, the research goals should be provided.
- There are inconsistencies in the reported data, which is a serious error. The authors should carefully check the entire manuscript to avoid similar mistakes. For example, the docking energy of site 1 is given as -24.3 kcal/mol in the main text but -23.4 kcal/mol in the legend of Figure 1. Similarly, the docking energy of site 2 is reported as -21 kcal/mol in the main text but -21.2 kcal/mol in the figure legend. Such inconsistencies must be carefully checked and corrected.
- The caption of Figure 1 is misleading: it refers to anti-PF4/PF4–integrin interaction, but the figure shows only PF4–integrin interactions without any antibody. This inconsistency must be corrected.
- All figures (Figures 1–6) are low resolution. The authors should provide high-resolution versions (≥300 DPI) in the revised submission.
- The sentence is unclear and the wording is inappropriate. The authors should carefully check the manuscript. For example, Lines 140-142: ‘To establish the specificity of PF4 binding to soluble integrins allbẞ3 (b) and avẞ3 (c), if ADAM15 disintegrin...’
- In the caption for Figures 3a and 3b, the sentence 'After washing, bound integrins were quantified using anti-ẞ3 (mAb AV10) and HRP-conjugated anti-mouse IgG' is repeated verbatim. This redundancy appears to be a copy-and-paste error and should be corrected.
Minor issues:
- Line 31: “suppressed” should be revised to “suppress”.
- Lines 39-41: The sentence “We previously discovered that the chemokine domain of pro-inflammatory chemokine CX3CL1 is a ligand for integrins avß3 and a4ß1 and bound to the classical ligand-binding site of integrins (site 1) (7),” should be corrected to “We previously discovered that the chemokine domain of pro-inflammatory chemokine CX3CL1 is a ligand for integrins avß3 and a4ß1 and binds to the classical ligand-binding site of integrins (site 1) (7).”
- Line 101: The word ‘activate’ should be in the singular form, ‘activates.’
- Line 102: The word ‘site’ should be in the plural form, ‘sites.’
- Line 139: There is a spelling error. The word ‘distintegrin’ should be ‘disintegrin.’
- Line 142: The word ‘suppress’ should be in the singular form, ‘suppresses’
- Section 2.6 title: ‘Plasma from healthy controls contain antibodies…,’ has a subject-verb agreement error. ‘contain’ should be revised to ‘contains.’
Author Response
This study identifies a new mechanism by which anti-PF4/PF4 complexes allosterically activate integrins, providing important insights into the pathogenesis of thrombotic thrombocytopenia and autoimmune diseases. The experiments are well designed, particularly the use of specific antibodies (RTO and KKO) and the site-directed mutant (4E) to examine the activation mechanism. The finding of RTO-like autoantibodies in healthy individuals is interesting and adds clinical relevance. However, several major and minor issues need to be addressed before publication.
Major issues:
I think the Introduction section is too brief. It lacks background on PF4 as an inhibitory chemokine, which the authors should add. Furthermore, the authors should clearly identify the research gap. Moreover, the research goals should be provided.
Response: PF4’s roles in the pathogenesis of diseases are so diverse, including angiostatic activity, that we included review articles. Research goals were provided.
There are inconsistencies in the reported data, which is a serious error. The authors should carefully check the entire manuscript to avoid similar mistakes. For example, the docking energy of site 1 is given as -24.3 kcal/mol in the main text but -23.4 kcal/mol in the legend of Figure 1. Similarly, the docking energy of site 2 is reported as -21 kcal/mol in the main text but -21.2 kcal/mol in the figure legend. Such inconsistencies must be carefully checked and corrected.
The caption of Figure 1 is misleading: it refers to anti-PF4/PF4–integrin interaction, but the figure shows only PF4–integrin interactions without any antibody. This inconsistency must be corrected.
All figures (Figures 1–6) are low resolution. The authors should provide high-resolution versions (≥300 DPI) in the revised submission.
Response: The inconsistency was corrected. Low resolution figures were replaced with high resolution figures.
The sentence is unclear and the wording is inappropriate. The authors should carefully check the manuscript. For example, Lines 140-142: ‘To establish the specificity of PF4 binding to soluble integrins allbẞ3 (b) and avẞ3 (c), if ADAM15 disintegrin…’
Response: The sentence was revised.
In the caption for Figures 3a and 3b, the sentence 'After washing, bound integrins were quantified using anti-ẞ3 (mAb AV10) and HRP-conjugated anti-mouse IgG' is repeated verbatim. This redundancy appears to be a copy-and-paste error and should be corrected.
Response: The error was corrected.
Minor issues:
Line 31: “suppressed” should be revised to “suppress”.
Lines 39-41: The sentence “We previously discovered that the chemokine domain of pro-inflammatory chemokine CX3CL1 is a ligand for integrins avß3 and a4ß1 and bound to the classical ligand-binding site of integrins (site 1) (7),” should be corrected to “We previously discovered that the chemokine domain of pro-inflammatory chemokine CX3CL1 is a ligand for integrins avß3 and a4ß1 and binds to the classical ligand-binding site of integrins (site 1) (7).”
Response: Corrected.
Line 101: The word ‘activate’ should be in the singular form, ‘activates.’
Line 102: The word ‘site’ should be in the plural form, ‘sites.’
Line 139: There is a spelling error. The word ‘distintegrin’ should be ‘disintegrin.’
Line 142: The word ‘suppress’ should be in the singular form, ‘suppresses’
Section 2.6 title: ‘Plasma from healthy controls contain antibodies…,’ has a subject-verb agreement error. ‘contain’ should be revised to ‘contains.’
Response: These errors were corrected.
Reviewer 2 Report
Comments and Suggestions for Authors
Takada et al. submitted a study entitled “Anti-PF4/PF4 complex induces allosteric activation of integrins αIIbβ3 and αvβ3, a potential mechanism of autoanti-PF4-induced thrombotic thrombocytopenia and autoimmune diseases.” The authors first performed molecular docking and identified that the PF4 monomer can bind to αvβ3, which was further confirmed by ELISA binding assays showing that PF4 can bind to αvβ3/αIIbβ3. They then found that PF4 alone could not induce integrin activation, whereas the PF4/RTO complex strongly activated integrins. Although the study demonstrates a degree of novelty, the current data are insufficient to support the authors’ conclusions, which severely limits the impact and extension of this work.
Major concerns:
- The title should not contain uncertain conclusions or speculative statements.
- Keywords are missing.
- Since the known targets of PF4 are αvβ3 and Mac-1, the authors need to explain how they came to hypothesize that PF4 may also bind to αIIbβ3 and activate it, and provide supporting data or rationale.
- The Introduction section repeats too much of the Results and needs to be revised. The authors are advised to seek assistance from a native English speaker for systematic language editing.
- Immunoprecipitation experiments are needed to further confirm the binding between PF4 and αvβ3/αIIbβ3.
- Both αIIbβ3 and PF4 are primarily expressed in platelets. Can they interact in resting platelets? What happens upon platelet activation? The authors should provide a more in-depth discussion of the cellular localization of PF4 with respect to αvβ3/αIIbβ3.
- The finding that PF4 alone does not activate integrins but the PF4/RTO complex does is interesting; however, the current data are insufficient to substantiate this conclusion. The authors should provide more detailed supporting evidence.
- The image quality is very poor. Please provide figures of at least 300 PPIresolution.
- Although this study preliminarily confirms that PF4 can bind to αvβ3 (which is already known), the current data do not convincingly support PF4 binding to αIIbβ3, and no functional studies were performed to clarify the biological consequences of PF4 binding to αvβ3/αIIbβ3. This limits the significance and expansion of the study.
- The authors mainly used the PF4 monomer for molecular docking, whereas PF4 functions physiologically as a tetramer. Therefore, the docking results have limited relevance. It is recommended that the authors also perform docking with PF4 tetramers to assess whether they can bind to αvβ3/αIIbβ3.
Based on the above issues, I recommend major revision.
Author Response
Comments and Suggestions for Authors
Takada et al. submitted a study entitled “Anti-PF4/PF4 complex induces allosteric activation of integrins αIIbβ3 and αvβ3, a potential mechanism of autoanti-PF4-induced thrombotic thrombocytopenia and autoimmune diseases.” The authors first performed molecular docking and identified that the PF4 monomer can bind to αvβ3, which was further confirmed by ELISA binding assays showing that PF4 can bind to αvβ3/αIIbβ3. They then found that PF4 alone could not induce integrin activation, whereas the PF4/RTO complex strongly activated integrins. Although the study demonstrates a degree of novelty, the current data are insufficient to support the authors’ conclusions, which severely limits the impact and extension of this work.
Major concerns:
The title should not contain uncertain conclusions or speculative statements.
Response: Title was revised.
Keywords are missing.
Response: Keywords were added.
Since the known targets of PF4 are αvβ3 and Mac-1, the authors need to explain how they came to hypothesize that PF4 may also bind to αIIbβ3 and activate it, and provide supporting data or rationale.
Response:PF4 is stored in platelet granules and released from platelets upon platelet degradation or activation. We previously showed that several cytokines that are stored in platelet granules bind to and activates αIIbβ3. This will lead to platelet activation and thrombosis. It is unclear if PF4 binds to αIIbβ3 and thereby induces platelet activation and thrombosis. (we added this statement)
The Introduction section repeats too much of the Results and needs to be revised.
Response: The description of the results in Introduction was moved to “Conclusion section in the end.
The authors are advised to seek assistance from a native English speaker for systematic language editing.
Immunoprecipitation experiments are needed to further confirm the binding between PF4 and αvβ3/αIIbβ3.
Response: Integrins have two binding sites (site 1 and site 2) and immunoprecipitation does not distinguish binding to site 1 or site 2. Instead, we have done mutations in the predicted interface to site 2 inhibit PF4 binding to site 2, and this is more precisely show the binding to site 2. In addition, we showed ivermectin, a site 2 antagonist, inhibited PF4/RTO-induced integrin activation.
Both αIIbβ3 and PF4 are primarily expressed in platelets. Can they interact in resting platelets? What happens upon platelet activation? The authors should provide a more in-depth discussion of the cellular localization of PF4 with respect to αvβ3/αIIbβ3.
Response: Although PF4 levels are very low in normal plasma, when platelets are degraded or activated, PF4 is released from granules and plasma PF4 levels dramatically increased (x1000). PF4 is activated by auto anti-PF4 in plasma (possibly PF4 tetramer is converted to monomer) and binds to site 2 of platelet αIIbβ3 and activates this integrin. Activated αIIbβ3 binds to plasma fibrinogen and induces platelet aggregation. (this statement was added to the end of Discussion)
The finding that PF4 alone does not activate integrins but the PF4/RTO complex does is interesting; however, the current data are insufficient to substantiate this conclusion. The authors should provide more detailed supporting evidence.
Response: We showed that PF4 mutant (4E) in the predicted site 2-binding interface is defective in activating integrins and act as an antagonist. This is best we can show now. We do not have reagents to show further evidence. We found the K50E mutant, which has been previously reported as a monomer, acted like wt PF4. Therefore, we are not sure if RTO converted PF4 tetramer to monomer at this point.
The image quality is very poor. Please provide figures of at least 300 PPI resolution.
Response: Low-resolution figures were replaced with the higher resolution figures.
Although this study preliminarily confirms that PF4 can bind to αvβ3 (which is already known), the current data do not convincingly support PF4 binding to αIIbβ3, and no functional studies were performed to clarify the biological consequences of PF4 binding to αvβ3/αIIbβ3. This limits the significance and expansion of the study.
Response:
We recently published that the binding of pro-inflammatory chemokines induce integrin activation (see Introduction). We found that human plasma contains RTO-like autoantibody in plasma. We can probably test if mouse PF4 (4E) mutant suppress PF4 in mouse in near future.
The authors mainly used the PF4 monomer for molecular docking, whereas PF4 functions physiologically as a tetramer. Therefore, the docking results have limited relevance. It is recommended that the authors also perform docking with PF4 tetramers to assess whether they can bind to αvβ3/αIIbβ3.
Response: The docking program Autodock3 was designed for docking of small molecules to their receptors. We found that the program can handle large ligand protein (e.g., 200 amino acids). However, PF4 tetramer (70 x 4=280 amino acids) is too big and Autodock could not handle it.
Reviewer 3 Report
Comments and Suggestions for Authors
This manuscript can be invaluable in the field of understanding PF4 mediated autoimmune diseases and help find potential interventions for such pathological conditions. However, the manuscript needs to address the comments below for it to become suitable for publication.
General comments:
- Too many self-citations. Please include other relevant references which support your previous findings – since the previous findings have been used as references to explain some of the current findings.
- The references start from #5 instead of #1 in the main text beginning in the Introduction section.
- Conclusion is missing.
- Mass spectrometry analysis and sequencing experimental missing.
- Please upload high resolution images for all the figures in the main text as well as the Supporting Information. Current images are hard to zoom into and look at.
Manuscript specific comments:
- Are these (αvβ3 and αIIbβ3) the only integrin heterodimers that these chemokines bind to?
- Line 54: Please include other relevant references. Right now, this is a self-citation.
- Line 89: Please add references.
- Since PF4 can bind integrins both the inactive and active state, once PF4 binds to the allosteric site (Site 2) upon activation in presence of Mn2+ activates integrins – can another molecule of PF4 bind to the Site 1?
- Line 116: Should the sentence not read: “….soluble integrins is not due to abnormality of our PF4 preparations”?
- Figure 2b: Why does the value for αvβ3 looks so low compared to αIIbβ3 in Fig 2b at PF4 concentration of 6.25 ug/mL compared to Fig 2a, where both of them looks superimposed at PF4 concentration of 10 ug/mL or less?
- Please also explain the use of several different concentrations of PF4 used for the different experiments.
- Figure 2e and 2f: In this experiment, you use DMSO as the control where there are no antagonists, while "None" describes no PF4 either. Is that correct? Were the soluble integrins premixed with the antagonists rather than mixing everything together into the PF4 immobilized wells? Additionally, we see activation of the integrins in the presence of antagonists. Why is that?
- Figure 2g and 2h: αvβ3: EDTA and CaCl2 also looks statistically significant compared to "None" as they have very close values to MnCl2. αIIbβ3: Same goes for CaCl2 and MgCl2. So, the statement "Mn2+ (1 mM) supported PF4 binding to site 1 well but other cations did not. " is not fully correct.
- Why are the mAbs (RTO or KKO at 10 ug/mL) used at 10x concentration to PF4 (1 ug/mL)? Can the presence of RTO itself activate the integrins? These controls are missing in Figure 3a and 3b.
- For generating mutant PF4, why was Glu substitutions chosen over Ala? By replacing Arg and Lys with Glu, you are also modifying the overall charge of PF4. Additionally, Glu still can act as a H-bond acceptor at physiological pH.
- Why was pooled plasma from the diseased controls (RA or SLE) not analyzed alongside the healthy ones for detecting the RTO-like autoantibodies?
- Is there binding data for NRG1 and IVM to the integrins available? Please refer to them if they are. Please include that in the discussion to show how either of them prevents binding of PF4/RTO complex to the integrins.
Author Response
This manuscript can be invaluable in the field of understanding PF4 mediated autoimmune diseases and help find potential interventions for such pathological conditions.
Response: Thank you for your kind comment.
However, the manuscript needs to address the comments below for it to become suitable for publication.
General comments:
Too many self-citations. Please include other relevant references which support your previous findings – since the previous findings have been used as references to explain some of the current findings.
The references start from #5 instead of #1 in the main text beginning in the Introduction section.
Response: Allosteric activation by binding to site 2 by inflammatory cytokines was originally discovered in my lab, and unfortunately only one lab (reference 11) followed this discovery. So, there are no related publications from other labs at this point. Thus, we need to cite our papers to explain the background of site 2. We hope other labs will study site 2 eventually but it will take some more time.
Conclusion is missing.
Response: Conclusion was added.
Mass spectrometry analysis and sequencing experimental missing.
Response: Experimental was added.
Please upload high resolution images for all the figures in the main text as well as the Supporting Information. Current images are hard to zoom into and look at.
Response: high-resolution figures were added (including supporting figure).
Manuscript specific comments:
Are these (αvβ3 and αIIbβ3) the only integrin heterodimers that these chemokines bind to?
Response: Several inflammatory chemokines bind to integrins other than αvβ3 and αIIbβ3 in our preliminary studies. We plan to publish the data in near future.
Line 54: Please include other relevant references. Right now, this is a self-citation.
Response: Allosteric activation by binding to site 2 by inflammatory cytokines was originally discovered in my lab, and unfortunately only one lab (reference 11) followed this discovery. So, there are no related publications from other labs at this point. Thus, we need to cite our papers to explain the background of site 2. We hope other labs will study site 2 eventually but it will take some more time.
Line 89: Please add references.
Since PF4 can bind integrins both the inactive and active state, once PF4 binds to the allosteric site (Site 2) upon activation in presence of Mn2+ activates integrins – can another molecule of PF4 bind to the Site 1?
Response: We believe that site 2 is open and site 1 is closed in high [Ca2+] environment like plasma. We observed that another integrin ligand (sPLA2-IIA) binds to site 2 and integrins are activated at low sPLA2-IIA concentrations and binding of FITC-labeled fibronectin too site 1 increases. However, with the sPLA2-IIA levels increase, site 2 is saturated and sPLA2-IIA starts to competes with FITC-labeled fibronectin. So, we see that FITC-fibronectin binding decreases.
Line 116: Should the sentence not read: “….soluble integrins is not due to abnormality of our PF4 preparations”?
Response: Corrected. Thanks.
Figure 2b: Why does the value for αvβ3 looks so low compared to αIIbβ3 in Fig 2b at PF4 concentration of 6.25 ug/mL compared to Fig 2a, where both of them looks superimposed at PF4 concentration of 10 ug/mL or less?
Response: We recently found that commercial soluble avb3 is unstable compared to soluble aIIbb3 for some reason we do not know. If we use more avb3 for activation or binding assays, we get stronger signals. So, the results are useful if we do not compare avb3 and aIIbb3.
Please also explain the use of several different concentrations of PF4 used for the different experiments.
Response: We determine the concentrations needed by titrating the PF4 for binding or activation. Binding and activation require different concentrations of PF4.
Figure 2e and 2f: In this experiment, you use DMSO as the control where there are no antagonists, while "None" describes no PF4 either. Is that correct?
Response: It is correct.
Were the soluble integrins premixed with the antagonists rather than mixing everything together into the PF4 immobilized wells?
Response: Antagonists were mixed with soluble integrins first and then added to the well (this statement was added to the legend.
Additionally, we see activation of the integrins in the presence of antagonists. Why is that?
Response: The assay buffer contains 1 mM Mn2+ to activate soluble integrins. This is why integrins were activated in the presence of antagonists.
Figure 2g and 2h: αvβ3: EDTA and CaCl2 also looks statistically significant compared to "None" as they have very close values to MnCl2. αIIbβ3: Same goes for CaCl2 and MgCl2. So, the statement "Mn2+ (1 mM) supported PF4 binding to site 1 well but other cations did not. " is not fully correct.
Response: description was corrected. “Mn2+ was most effective and other cations are less active”.
Why are the mAbs (RTO or KKO at 10 ug/mL) used at 10x concentration to PF4 (1 ug/mL)?
Response: mAbs=150K and PF4 is 8K (70 amino acids), and the numbers of molecules are comparable.
Can the presence of RTO itself activate the integrins? These controls are missing in Figure 3a and 3b.
Response: KKO or mouse IgG (as negative controls) did not activate integrins.
For generating mutant PF4, why was Glu substitutions chosen over Ala? By replacing Arg and Lys with Glu, you are also modifying the overall charge of PF4. Additionally, Glu still can act as a H-bond acceptor at physiological pH.
Response: Arg/Lys usually play critical roles in integrin binding. Arg/Lys to Glu mutation is “charge-reversal mutations and is widely used. In our experience, this mutation is more effective than Arg/Lys to Ala mutations in blocking integrin binding.
Why was pooled plasma from the diseased controls (RA or SLE) not analyzed alongside the healthy ones for detecting the RTO-like autoantibodies?
Response: The experiments were not done at the same time. Patient sera from patients were tested first, since we believed that RTO-like activity is present in sera. We then found that plasma, not sera, contains RTO-like activity. We always included RTO as a positive control.
Is there binding data for NRG1 and IVM to the integrins available? Please refer to them if they are. Please include that in the discussion to show how either of them prevents binding of PF4/RTO complex to the integrins.
Response: The paper came out recently. We included the reference 26.
NRG1 and IVM bind to site 2 but act as site 2 antagonists. They compete with PF4/RTO complex for binding to site 2.
Round 2
Reviewer 1 Report
Comments and Suggestions for Authors
Thank you for your revised manuscript and for carefully addressing the previous comments. All major and minor concerns have been satisfactorily resolved, and I recommend acceptance.
Author Response
Thank you.
Reviewer 2 Report
Comments and Suggestions for Authors
The authors have submitted a revised manuscript, which has addressed certain issues; however, the most critical concerns remain unresolved:
-
Co-immunoprecipitation (Co-IP) experiments are essential in this study. Without them, it is difficult to ascertain the binding of PF4 to αvβ3/αIIbβ3. Although Co-IP cannot determine the exact binding sites, it can substantively confirm their binding capability. This evidence is crucial for the present study; otherwise, the conclusions drawn cannot be convincingly supported.
-
Given that Autodock3 was originally designed for docking small molecules with their receptors and is limited in handling PF4 oligomer binding, I believe the authors should employ alternative approaches for molecular docking. This would help clarify whether the binding sites of PF4 oligomers differ from those of monomers, thereby providing a more in-depth explanation of their functional implications.
Unless the authors can adequately address the above issues, I cannot recommend this study for publication.
Author Response
-
Co-immunoprecipitation (Co-IP) experiments are essential in this study. Without them, it is difficult to ascertain the binding of PF4 to αvβ3/αIIbβ3. Although Co-IP cannot determine the exact binding sites, it can substantively confirm their binding capability. This evidence is crucial for the present study; otherwise, the conclusions drawn cannot be convincingly supported.
Response: We respectfully disagree with the comment. We have done ELISA-type binding assays in Fig. 1. We have immobilized PF4 to wells and blocked the wells with BSA. We then incubated wells with soluble integrins and detected the bound integrins with antibodies specific to human beta 3. Furthermore, we showed that the PF4-integrin interaction is affected by cations, antagonists to integrins, and inhibited by known integrin ligands. This experiment (ELISA) is essentially the same as immunoprecipitation and probably more quantitative than immunoprecipitation. In ELISA-type integrin activation assays, we used anti-PF4 antibody (RTO) to convert PF4 to active form. We used anti-beta3 to detect soluble integrins that bound to immobilized fibrinogen fragment. ELISA is again essentially the same as immunoprecipitation and is more quantitative than immunoprecipitation. We believe that immunoprecipitation would not provide additional information.
2. Given that Autodock3 was originally designed for docking small molecules with their receptors and is limited in handling PF4 oligomer binding, I believe the authors should employ alternative approaches for molecular docking. This would help clarify whether the binding sites of PF4 oligomers differ from those of monomers, thereby providing a more in-depth explanation of their functional implications.
Response: It would not be possible for us to use alternative programs for molecular docking in the limited time frame of this revision (10 days). Since we know that soluble PF4 tetramer does not bind to site 2 and activate integrins in binding and activation assays, we believe that docking simulation predict that PF4 tetramer shows little of no affinity to integrins (based on docking energy). Immobilized PF4 show integrin binding, but this may be because immobilization modifies the PF4 conformation and exposes integrin binding sites of PF4. In future experiments, we will need to show if antibodies (RTO or autoantibody) convert PF4 tetramer to active monomer.
Since point mutations (4E) of site 2 binding interface inhibit the RTO/PF4-induced integrin activation, we predict that monomer accesses site 2, but tetramer does not due to steric hindrance. We do not expect that docking simulation of tetramer identifies another site 2 binding site.
Reviewer 3 Report
Comments and Suggestions for Authors
The authors have addressed the concerns in a satisfactory manner.
Author Response
Thank you.
Round 3
Reviewer 2 Report
Comments and Suggestions for Authors
- I still suggest that the author should include the results of the Co-IP, as this is crucial for enhancing the credibility of this research.
- I think it is more than sufficient to complete the molecular docking within 10 days. Whether the PF4 polymer can bind to the integrin is also of great significance for this study. If the PF4 polymer cannot bind to the integrin, it obviously can enhance the significance of this study. Please provide this result.
Author Response
- I still suggest that the author should include the results of the Co-IP, as this is crucial for enhancing the credibility of this research.
Response: If anti-PF4 (RTO) is already present in the assay system, and co-PI with anti-integrin antibody is not possible. If anti-PF4 is not present, co-PI with anti-integrin will not precipitate PF4, since PF4 does not interact with integrins.
Using ELISA-type assay, we showed that integrins bind to immobilized fibrinogen fragment in the presence of anti-PF4 (RTO), but not KKO. This suggests that RTO induce PF4 binding to integrins. We used anti-human b3 antibody (AV10) to detect bound integrins, suggesting that RTO-induces PF4-integrin interaction. co-IP will not allow such analysis.
2. I think it is more than sufficient to complete the molecular docking within 10 days. Whether the PF4 polymer can bind to the integrin is also of great significance for this study. If the PF4 polymer cannot bind to the integrin, it obviously can enhance the significance of this study. Please provide this result.
Response: We have not seen docking of large protein ligands (280 amino acids) to integrins in the literature. We need to find which other docking programs allow such docking. We do not expect to find any. Even such program exists, setting up another docking systems may take time and will encounter other problems. Alternatively, we superposed the PF4 tetramer structure to PF4 monomer/integrin complex. This analysis suggests that there may be some steric hindrance. But we are not 100% sure. This is the best we could do at this point. I hope you understand the situation.